# Difference between the North Atlantic and Pacific meridional overturning circulation in response to the uplift of the Tibetan Plateau

Baohuang Su[1, 4], Dabang Jiang[1, 2, 3, 4], Ran Zhang[1], Pierre Sepulchre[5], and Gilles Ramstein[5]

[1]Institute of Atmospheric Physics, Chinese Academy of Sciences, Beijing 100029, China

[2]Joint Laboratory for Climate and Environmental Change at Chengdu University of Information Technology, Chengdu 610225, China

[3]CAS Center for Excellence in Tibetan Plateau Earth Sciences, Beijing 100101, China

[4]University of Chinese Academy of Sciences, Beijing 100049, China

[5]Laboratoire des Sciences du Climatet de l'Environnement/IPSL, CEA-CNRS-UVSQ, UMR8212, Orme des Merisiers, CE Saclay, 91191 Gif-sur-Yvette Cedex, France

**Abstract:** The role of the Tibetan Plateau (TP) in maintaining the large-scale overturning circulation in the Atlantic and Pacific is investigated using a coupled atmosphere–ocean model. For the present day with a realistic topography, model simulation shows a strong Atlantic meridional overturning circulation (AMOC) but a near absence of the Pacific meridional overturning circulation (PMOC), which are in good agreement with the present observations. In contrast, the simulation without the TP depicts a collapsed AMOC and a strong PMOC that dominates deep water formation. The switch in deep water formation between the two basins results from changes in the large-scale atmospheric circulation and atmosphere–ocean feedback over the Atlantic and Pacific. The intensified westerly winds and increased freshwater flux over the North Atlantic cause an initial slowdown of the AMOC, while the weakened East Asian monsoon circulation and associated decreased freshwater flux over the North Pacific give rise to the initial intensification of the PMOC. The further decreased heat flux and the associated increase in sea-ice fraction promote

the final AMOC collapse over the Atlantic, while the further increased heat flux leads to the final

PMOC establishment over the Pacific. Although the simulations were done in a cold world, it still

importantly implicates that the uplift of the TP alone could have been a potential driver for the

reorganization of PMOC–AMOC between the Late Eocene and Early Oligocene.

## 1. Introduction

The uplift of the Tibetan Plateau (TP) was a major tectonic event that had occurred throughout

the Cenozoic, and its gradual growth had exerted a strong influence on the atmospheric circulation

and climate (Molnar et al., 2010). Since the pioneering work of Bolin (1950), the impacts of

mountain uplift on regional and global climate have been extensively investigated. Nevertheless,

most studies have emphasized the role of mountain ranges on atmosphere dynamics, while

quantifications of the associated impact on ocean dynamics have been rare. For example, most

previous works have taken atmospheric general circulation models to address regional climate

effects, notably the Asian monsoon and arid environment evolutions (e.g., Ruddiman and Kutzbach,

1989; Ramstein et al., 1997; An et al., 2001; Liu and Yin, 2002; Jiang et al., 2008; Zhang et al.,

2015). However, simulations have recently been applied to investigate the effect of mountain uplift

in the context of the atmosphere–ocean system, and a few studies have proposed that the uplift of

the Andes (Sepulchre et al., 2009) and Rocky Mountains (Seager et al., 2002) is closely linked to

the evolution of oceanic circulations, including the Gulf Stream and Humboldt Current, and the El

Niño–Southern Oscillation system (Feng and Poulsen, 2014). Although it has been indicated that

the TP uplift affects sea surface temperatures, sea surface salinity, precipitation, and trade winds for

both the Pacific and equatorial Indian Ocean (Abe et al., 2003; Kitoh, 2004; Okajima and Xie,

2007), the influence of the TP uplift on the high-latitude oceanic circulations, particularly in the

North Atlantic, has rarely been explored.

46         The potential importance of mountain uplift in modifying the oceanic thermohaline circulation

has previously been investigated. Ruddiman and Kutzbach (1989) indicated that mountain
uplift-induced changes in the North Atlantic surface circulation are expected to increase the North
Atlantic Deep Water formation. In addition, Rind et al. (1997) performed the coupled model
simulations with and without TP and proposed that the TP may have a considerable impact on the
large-scale meridional overturning circulation (MOC). However, the integration time used in this
pioneering simulations was too short to fully evaluate the deep oceanic circulation response, and
thus more studies are still needed to evaluate the possible role of the TP in modulating the MOC.

54         On a geological timescale, remarkable reorganization and evolution of the large-scale oceanic

overturning circulation, from the Southern Ocean deep water dominating mode to the modern-like
North Atlantic deep water mode, have been evidenced through the Late Eocene to the Early
Oligocene (Wright and Miller, 1993; Davies et al., 2001; Via and Thomas, 2006). This dramatic
shift is possibly associated with major rearrangements in the ocean seaways and other tectonic
changes, although the ultimate trigger is still being debated (Zhang et al., 2011). In addition, it is
suggested that the regional surface of the TP had reached a high elevation of more than 4000 meters
around 40 Ma ago (Dupont-Nivet et al., 2008; Wang et al., 2008), although debates regarding
paleoaltitude reconstructions remain (Botsyun et al., 2016). Given this timing of TP uplift, it is
important to quantify the contribution of TP uplift on the meridional oceanic circulation of the
Northern Hemisphere. In this study, therefore, two coupled atmosphere–ocean numerical
integrations, with and without TP, are designed to investigate the role of the TP on the Atlantic
MOC (AMOC) and Pacific MOC (PMOC).

## 2. Model, experimental design, and density flux analysis

### 2.1. Model and experiments

The Community Earth System Model (CESM) version 1.0.5 of the National Center for Atmospheric Research is a widely used, well-validated coupled model with dynamic atmosphere, land, ocean, and sea-ice components (Gent et al., 2011). It is applied to this study at a low-resolution configuration that is computationally efficient and well-described (Shields et al., 2012) and employs an atmospheric horizontal grid of roughly $3.75° \times 3.75°$ (T31) with 26 vertical levels. The ocean model adopts a finer oceanic horizontal grid, with a nominal $3°$ resolution increasing to $1°$ near the equator ($116 \times 100$ grid points, latitude by longitude) and 60 unevenly spaced layers in the vertical direction. The sea-ice and land models share the same horizontal grids as the ocean and atmosphere models, respectively, where the sea-ice component is a dynamic–thermodynamic model that includes a subgrid-scale ice thickness distribution and energy-conserving thermodynamics (Holland et al., 2012).

Two experiments are conducted; firstly, a control run with the modern topography (MTP, Figure 1a), and secondly a sensitivity run where topography within the region of 20–60°N and 60–140°E at altitudes higher than 200 m is set to 200 m (NTP, Figure 1b), which enables examination of the climate effect in relation to the TP topography. This TP uplift configuration has been referred to in the majority of previous simulation works (e.g., Liu and Yin, 2002; Jiang et al., 2008). This greatly simplified topographic setting is not intended to represent a realistic scenario constrained by the geological evidence and instead represents two end-members of the potential growth histories of the TP. So, it is important to note that these experiments only aim to investigate the TP uplift occurring in "a cold world" with an atmospheric $CO_2$ corresponding to the pre-industrial values (284.0 ppm). With the exception of topography, all the other boundaries, such

as land–sea distribution and orbital parameters, are prescribed to pre-industrial conditions. The MTP
is continually integrated for 1100 years, and the NTP is additionally integrated for another 1840
years starting from the year 1100 of the MTP. Global mean surface air temperature and sea
temperature at a depth of 1000 m are shown in Figure 1c. Both simulations reach equilibrium states
after more than 1000 model years of integration time, and the final 200 years of both cases are
applied for our climate state analysis.
2.2. Density flux analysis
Because one of the major aims of this paper is to analyze changes in the meridional oceanic
circulation, we decided to focus on the density flux parameter, which is appropriate to diagnose
these oceanic circulation changes. The dense deep water masses are formed in the area with
relatively high surface density achieved by cooling or increasing salinity. To better understand
which processes dominated the MOC changes in simulations, it is instructive to further analyze the
time evolution of density fluxes budget. Therefore, a density flux analysis method, in which the
total density flux decomposes into the haline contribution due to freshwater flux and the thermal
contribution due to heat flux (Schmitt et al., 1989), is adopted in our study. The total density flux is
calculated from a linearized state equation of seawater, as
$$F_\rho = -\alpha \cdot \frac{Q}{C_p} + \rho(0, T) \cdot \beta \cdot \frac{(E - P - R - I) \cdot S}{1 - S}$$

$F_\rho$ is the total density flux, $-\alpha \cdot \frac{Q}{C_p}$ is thermal density flux, and $\rho(0, T) \cdot \beta \cdot \frac{(E-P-R-I) \cdot S}{1-S}$ is haline
density term. $C_p$, $T$, and $S$ are the specific heat capacity, surface temperature and salinity of
seawater, respectively. $\alpha$ and $\beta$ are the thermal expansion and haline contraction coefficients,
respectively. $\rho(0, T)$ is the density of freshwater with a salinity of 0 psu and temperature of $T$. $Q$
represents the net surface heat flux. $E$, $P$, $R$, and $I$ denote the freshwater fluxes due to
evaporation, precipitation, river runoff, and sea-ice melting (or brine rejection), respectively.
## 3. Results
### 3.1. Changes in AMOC and PMOC
There are evident changes in the AMOC and PMOC indices in response to the TP uplift
(Figure 1d). With MTP, the AMOC stabilizes at around 17 Sv (Sv = $10^6$ m$^3$ s$^{-1}$) for more than 1000
years (Figure 1d, 1–1100 years, red line), which agrees with the observations 18.7 $\pm$ 5.6 Sv for
2004–2005 (Cunningham et al., 2007), but with NTP there is a continual weakening of the AMOC
until the point of quasi-collapse (ca. 2 Sv, Figure 1d). In contrast, the PMOC of NTP begins at a
sluggish level from MTP (Figure 1d, 1–1100 years, purple line) and takes as long as 1200 years to
reach an equilibrium state that is comparable to the level of the AMOC in MTP (ca. 18 Sv, Figure
1d, 1101–2940 years, purple line). In agreement with the dramatic responses of AMOC and PMOC,
sea surface salinity increases in the North Pacific but decreases in a broad area of the North Atlantic
(Figure 2b). To fully understand the different behaviors between AMOC and PMOC in NTP, in the
following sections we further analyze changes in the atmospheric and oceanic circulations and the
atmosphere–ocean feedbacks.
### 3.2. Atmospheric responses
The modified AMOC and PMOC are linked to the large-scale atmospheric circulation changes.
In terms of model results of the NTP relative to the MTP, the surface air temperature over and
around the TP and in the North Pacific increased but decreased over the North Atlantic (Figure 2a),
which agrees with the previous simulations (Broccoli and Manabe, 1992; Kutzbach et al., 1993). In
addition, there are intensified westerlies over the North Atlantic and weakened subtropical
anticyclones and trade winds over the North Pacific (Figure 2c); the former results from a

significant increase in the meridional pressure gradient driven by a large-scale equatorward shift of air mass occupying the current position of the TP (Figure 2c) and from a reduced drag of the orographically induced gravity waves associated with the absence of the TP (Palmer et al., 1986; Sinha et al., 2012), while the latter is derived from the weakening of zonal Eurasia–Pacific thermal contrast in the middle troposphere (not shown), especially in boreal summertime, in relation to removal of the mountains (Ruddiman and Kutzbach, 1989; Rodwell and Hoskins, 2001; Kitoh, 2004).

When the TP is removed, the atmospheric moisture transport between the Pacific and Atlantic Oceans undergoes a basin-basin asymmetric redistribution as a response to the large-scale circulation anomalies (Figure 2c). In comparison with MTP results, the NTP simulation shows large amounts of anomalous westerly moisture flux transported through the lowlands of Central America to the North Atlantic, causing weak moisture convergence therein (Figure 2d). In addition, removal of the TP leads to a significant divergence of moisture over East Asia and the western North Pacific marginal seas (Figure 2d), which is linked to a weakened monsoon circulation and is consistent with the previous simulations using both atmospheric and coupled ocean–atmosphere general circulation models (Liu and Yin, 2002; Kitoh, 2004; Molnar et al., 2010). The anomalies of atmospheric circulation shown above are derived partially from the positive feedback caused by the changed sea surface temperature due to the removal of the TP. In particular, the weakening of both the Asian monsoon and North Pacific subtropical anticyclone in association with the TP removal are shown to be greater in the context of coupled model as compared to that in atmosphere-only model due to the additional ocean–atmosphere feedback (Kitoh, 2004). Thus, the atmosphere–ocean feedbacks also play an important role in maintaining the inter-basin atmospheric moisture asymmetric redistribution.

3.3. Oceanic responses and atmosphere–ocean feedbacks
3.3.1. Changes in freshwater and sea-ice
The above changes in the large-scale atmospheric circulation markedly decrease the total
ocean density flux in the North Atlantic (Figure 3a, brown), supporting the trend of the AMOC
(Figure 1d). Both the increases of net freshwater and wind-driven sea-ice expansion are responsible
for the initial reduction of total ocean density flux and further induce a gradual weakening of the
AMOC. In more details, on the one hand, the anomalous atmospheric circulation associated with
the removal of the TP transports more water vapor northward (Figure 2d) over the North Atlantic
Ocean, causing more precipitation at the beginning of NTP simulation (Figure 3b, ca. 1101–1200
years, red line). Correspondingly, the net freshwater flux (precipitation plus runoff minus
evaporation) convergence into the North Atlantic basin at 40 °–70 °N increases by 0.005 Sv (~3%)
and 0.025 Sv (~16%) at the initial and final states of NTP simulations (Figure 3b, green),
respectively. It should be noted that these changes in freshwater budget in our simulations are less
than approximately 0.0446 Sv and 0.097 Sv of recent simulations with and without global
mountains (Maffre et al., 2017), and the time for a complete collapse of the AMOC in our NTP
simulation (Figure 3b) is also longer (approximately 700 years) than for their experiments without
global mountains (approximately 400 years). Such a difference should be related to the
experimental design and the sensitivity of the models to freshwater forcing.
There is also a significant increase in the area-averaged sea-ice coverage over the North
Atlantic through wind-driven processes (Figure 3c, green). With the TP, the annual mean sea-ice
forms mainly in the northern and western region of the sub-polar North Atlantic, and it shifts
southward and eastward when driven by cyclonic wind stress associated with the Icelandic Low,
and melts in the Labrador Sea (sub-polar gyre) caused by a warm condition (Figure 4a). By

comparison, after removal of the TP, anomalously intensified cyclonic winds induce an anomalous

eastward sea-ice velocity (Figure 4c) and also a rapid eastward shift of the sea-ice margin (Figure

4c). Meanwhile, the locally melted sea-ice due to thermodynamics processes reduces in the

southeast of Greenland (red shading, Fig. 4c), but increases in the south of Greenland (blue shading,

Fig. 4c). It suggests that there is more sea-ice transporting from the high latitudes into the sub-polar

gyre region, and the anomalous expansion of sea-ice margin in this region primarily originates from

the wind-driven eastward transportation (dynamics processes), but not the local formation

(thermodynamic processes). Because of this increased sea-ice through thermodynamically

insulating the sea water from the freezing air, the release of sensible and latent heat into the

atmosphere decreases and the density of sea water finally reduces, which processes have also been

previously elucidated by Zhu et al. (2014).

Moreover, the total ocean density flux increases in the North Pacific in response to the removal

of the TP. Due to the weakened Asian monsoon circulation and associated decrease in rainfall and

runoff after lowering the topography, the net freshwater flux received by the North Pacific decreases

by 0.08 Sv (~26%) and 0.12 Sv (~40%) during the initial and end stages of the NTP simulation,

respectively (Figure 6b, green). This continuous negative freshwater flux forcing tends to increase

density and initially leads to the formation of the North Pacific dense water, which is verified from

changes in the haline density flux (Figure 6a). Specifically, during the first 200 years of the NTP run,

the haline density flux constantly produces a net positive contribution to the total density relative to

the MTP haline term (Figure 6a, blue line). Meanwhile, the thermal density flux remains at a lower

level (Figure 6a, ca. 1101–1300 years, red line) relative to the MTP. Thus, it indicates that the

initially increased density of the North Pacific is largely attributed to the haline density term, but

not the thermal density term.

## 3.3.2. Roles of the atmosphere–ocean feedbacks

The aforementioned weakening of the AMOC due to the atmospheric processes further triggers a positive atmosphere–ocean feedback loop through reducing northward heat transport, and subsequent decreasing sea surface temperatures, then allowing sea-ice to expand, suppressing the release of evaporating latent and sensible heat, and reducing the sea water density, and further weakening the AMOC, as previously shown in Jayne and Marotzke (1999) and Zhu et al. (2014). Note that the negative effect of net freshwater becomes increasingly unimportant in comparison to the heat flux feedback associated with the latent/sensible heat changes (Figure 3a). Finally, the thermal density flux decreases by 49% relative to the MTP run, which substantially dominates the total density flux changes (Figure 3a). To be specific, the annual mean total density flux and mixed layer depth over the North Atlantic, especially around the Iceland where the collapse of deep water formation occurs, is dramatically decreased in NTP (Figure 5d, the maximum mixed layer depth is approximately 100 m) in comparison to that in MTP (Figure 5a, the maximum mixed layer depth is approximately 900 m). Moreover, this reduced total density flux over the North Atlantic is more attributed to the decreased thermal density flux associated with less latent and sensible release (Figure 5e) than the changed haline density flux (Figure 5f).

Atmosphere–ocean feedbacks also strengthen the PMOC. Due to the initial development of the PMOC mentioned in section 3.1, a positive feedback (as pointed out in Warren (1983)) is initiated by the intensifying meridional oceanic circulation, which transports warmer subtropical water northward and leads to the buoyancy loss and evaporation increase (Figure 6b). This feedback is also able to re-trigger PMOC enhancement. By comparison to the changes in the North Atlantic, both the regionally averaged sea-ice coverage (Figure 6c) and February sea-ice margin (Figure 4f) over the North Pacific experience a slightly northward retreat and have a relatively smaller effect on

the simulated strengthening of the PMOC. Over a longer time, the thermal density flux, which is
due to the loss of total heat, contributes more to the total density flux than the haline flux in relation
to reduction in the net freshwater discharge (Figure 6b). Spatially, both increased total density flux
and mixed layer depth in the North Pacific Ocean show opposite change characteristics with the
North Atlantic (Figure 7d). Correspondingly, in comparison to the MTP, there is a widespread
increase of the thermally induced density flux in the sub-polar North Pacific in NTP (Figure 7f), but
with little spatially changed in the haline density flux (Figure 7f). Thus, in contrast to the results
shown in the North Atlantic, the increased total heat exchange between the atmosphere and ocean
due to the processes of sensible and latent heat releases (Figure 6b) ultimately becomes a dominant
factor in maintaining a vigorous PMOC by controlling the increased total density flux (Figure 6a).
## 4. Conclusions and Discussion
This study investigates the effect of TP uplift on the large-scale oceanic circulation using a
low-resolution version of CESM. Results show that the removal of the TP initially changes the
wind-driven atmospheric moisture transport process and the wind-driven sea-ice coverage
expansion process, which are responsible for the initial weakening of the AMOC. Meanwhile, the
suppressed monsoonal circulation in East Asia and the western Pacific marginal seas induces the
decrease of rainfall and runoff and further causes the initially increased PMOC. Moreover, the
positive feedback further changes the AMOC and PMOC. In particular, the AMOC weakening can
further decrease the North Atlantic sea surface temperatures, ocean–atmosphere temperature
contrast, evaporation, and precipitation, and subsequently increase sea-ice coverage. These
processes together cause the final changes of the AMOC and PMOC (Figure 8).
A previous study demonstrated the role of Rocky Mountain uplift on heat transport and Gulf

Stream patterns in the North Atlantic (Seager et al., 2002). In this study, we focus on the most prominent long-term orogenesis occurring since the Eocene: the TP and Himalayan uplift and associated impacts on the MOC. Our results can be compared with those derived from the earlier simulations, although experimental configurations differ somewhat. It has been indicated that the removal of global mountains triggers the collapse of deep water in the North Atlantic but enables formation in the North Pacific in three different coupled models (Schmittner et al., 2011; Sinha et al., 2012; Maffre et al., 2017). The simulated weakening of the AMOC is also qualitatively consistent with recent experiments using a decreased elevation of the TP and Central Asia (Fallah et al., 2016). However, only TP topography is reduced in our study, but our results are comparable with those of the earlier studies, therefore highlighting the key role that TP has played in forming the current large-scale deep oceanic circulation pattern. Nevertheless, given that all existing simulations (including ours) have used a rather coarse resolution of the coupled model configuration, it is considered that a finer resolution model may provide a better representation of the western boundary currents and allow for a more accurate and realistic resolving of the ocean eddies, which are believed to be critically important oceanic processes that should be taken in the realistic simulations of the AMOC (Spence et al., 2008). In addition, the low-resolution CESM is also found to generally have a cold bias with the underestimated ocean heat transport and excessive Arctic sea-ice (Shields et al., 2012), which could potentially exert modulations on the AMOC weakening. It is thus considered that investigating the response of the PMOC and AMOC to the TP uplift using an atmosphere–ocean general circulation model with a higher spatial resolution would be useful. Besides, the robust changes in the AMOC and PMOC and the associated mechanisms due to the TP uplift can be evaluated through multi-model comparison.

Based on a comprehensive analysis of modern climatological data, Warren (1983) and

Emile-Geay et al. (2003) hypothesized that the present MOC (mainly occurring in the Atlantic but
not in the Pacific) is determined by the large mountains, namely the Himalayas and Rockies, which
induce an asymmetric distribution of wind stress and moisture transport features between the
Atlantic and Pacific basins. However, previous studies have also demonstrated that the asymmetric
continental extents and basin widths (basin geometries) between the two basins (Weaver et al., 1999;
Nilsson et al., 2013) also play a possible key role in maintaining the present day AMOC. Our
simulations support the hypothesis proposed by Warren (1983) and highlight the significant role of
the TP alone in maintaining the modern AMOC. Moreover, the similar PMOC–AMOC seesaw
dynamics have also been seen in simulations (Saenko et al., 2004; Chikamoto et al., 2012; Hu et al.,
2012) as well as in reconstructions (Okazaki et al., 2010; Menviel et al., 2014; Freeman et al., 2015)
for the last deglaciation. Such studies have also suggested that large PMOC–AMOC seesaw
modulations can be triggered by slight changes in the freshwater/salinity redistributions between the
Pacific and Atlantic. Furthermore, we provide an insight that the maintenance mechanism of PMOC
in the simulation without the TP, to some extent, is the same as the AMOC in the present day.
Specially, for the current North Atlantic, there is a persistently northward movement of warm and
salty water mass from the tropical-subtropical Gulf Stream region into North Atlantic and farther
poleward into the Norwegian and Greenland Seas, where it is exposed to very cold atmospheric
temperatures and followed by a gradual cooling and in turn a higher density due to the release
substantial sensible and latent heat into the overlying cold atmosphere, which is the same as PMOC,
before eventually forming the North Atlantic Deep Water.

291         Our simulations have potential implications for understanding paleotemperature reconstructions

and paleoceanographic circulation reorganization. The Earth has experienced a long-term cooling
trend throughout the Cenozoic as testified by many proxies and stacked records (Zachos et al., 2001,

2008), in association with an increased equator-to-pole thermal gradient. A very important contribution to understanding the large cooling during the Cenozoic has been determined as the drastic decrease in atmospheric $CO_2$ since the Eocene (DeConto and Pollard, 2003; DeConto et al., 2008). On the other hand, a study with new data base further indicated that this thermal evolution has been different among ocean basins during the Cenozoic (Cramer et al., 2009), and this differing evolutionary pattern between basins is largely related to the large-scale ocean dynamics and tectonic events (Zhang et al., 2011). Moreover, epsilon-Neodymium (eps-Nd) isotopes in the deep Pacific suggest that the North Pacific was characterized by vigorous deep water formation during ca. 65–40 Ma (Thomas, 2004). Other new eps-Nd records also confirm that the overturning circulation was already established in the high-latitude North Pacific prior to 40 Ma (Hague et al., 2012; Thomas et al., 2014). Comparatively, a modern-like bipolar oceanic circulation, characterized by two branches of deep water formation in the Southern Ocean and the North Atlantic, began in the late Eocene (~38.5 Ma) in relation to the effect of Southern Ocean gateway openings (Borrelli et al., 2014). Several records also support that the onset of the present AMOC state began at the Eocene–Oligocene transition (~34 Ma) in association with the tectonic deepening of the Greenland–Norwegian Sea (Wright and Miller, 1993; Davies et al., 2001; Via and Thomas, 2006). However, it is likely that the intermittent Cenozoic uplift of the TP reached a certain height by the Early Oligocene, as shown in geologic evidences (Dupont-Nivet et al., 2008; Wang et al., 2008). Our own contribution demonstrates that the major uplift occurring during this period was also an important player in climate changes via hydrologic and ocean dynamics changes. Indeed, we pinpoint the drastic effect of TP uplift alone on the distribution of the northern hemispheric MOCs and potentially provide clues for proxy record interpretation.

Finally, this simulation is performed with the constant atmospheric $CO_2$ concentrations at the

pre-industrial, whereas it was higher during the uplift phase in the real world. In the context of past
warm world, such as Late Eocene, the climate conditions are accompanied by high atmospheric
$CO_2$ concentration, limited sea ice extent, and significantly modified land–sea distribution. Under
these warmer boundary conditions, the responses of AMOC to the TP induced freshwater forcing
may be very different from the modern conditions (e.g., Vavrus and Kutzbach, 2002). Therefore, it
will be necessary to perform further numerical experiments with more realistic boundary conditions
to accurately investigate the contribution of the TP uplift on the oceanic circulation and therefore to
be able to compare with data reconstructions.
*Acknowledgements* We sincerely thank the two anonymous reviewers for their insightful
comments and suggestions to improve this manuscript. We also thank Dr. Jiang Zhu for discussions
and technical support during the writing of the paper. This work was supported by the National
Natural Science Foundation of China (41421004, 41572159, and 41625018).

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

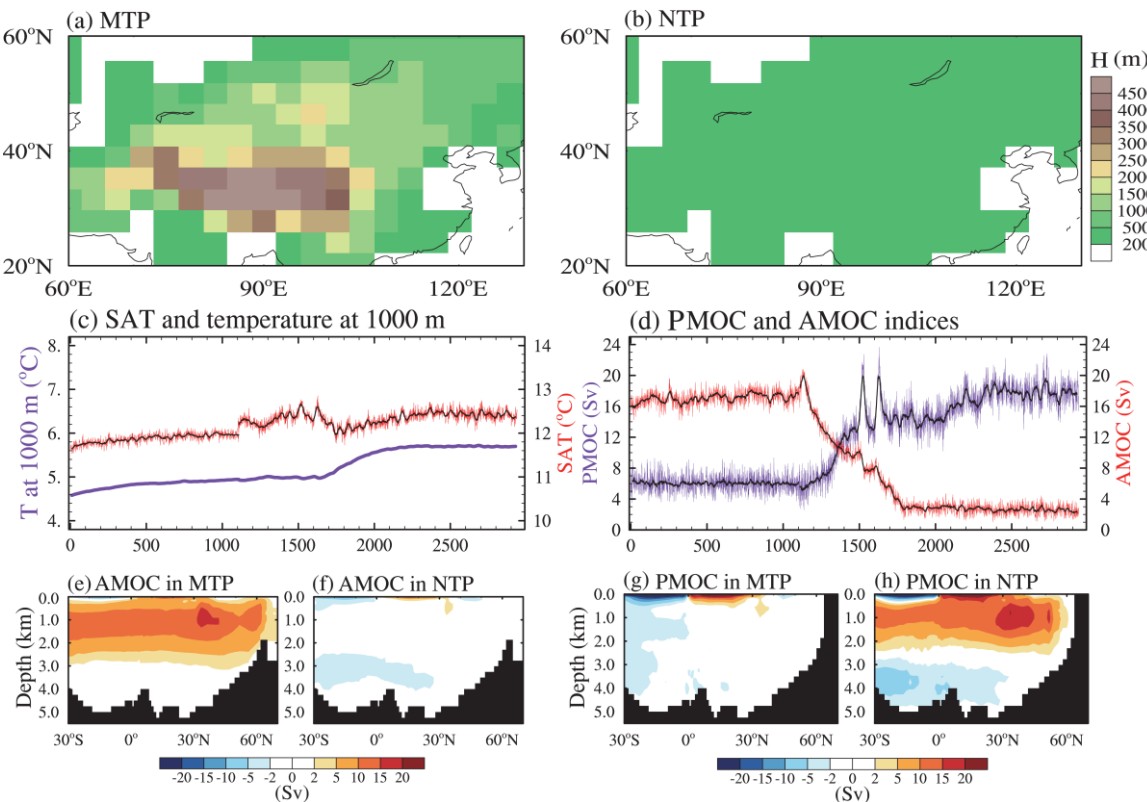

**Figure 1.** Two topographic height configurations used in experiments: (a) MTP and (b) NTP. (c) Time series of global mean annual 2-m surface air temperature (SAT) and sea temperature at 1000 m depth in MTP (1–1100 years) and NTP (1101–2940 years) simulations; bold black lines show 21-year running mean. (d) Same as (c) but for PMOC and AMOC indices, respectively. AMOC and PMOC indices are defined as the annual maximum of the meridional stream function value north of 28°N and below the depth of 500 m over the North Atlantic and Pacific, respectively. (e–h) Climatological annual mean Atlantic and Indian–Pacific meridional overturning stream function in MTP (e and g) and NTP (f and h); positive (negative) shading represents clockwise (counterclockwise) circulations.

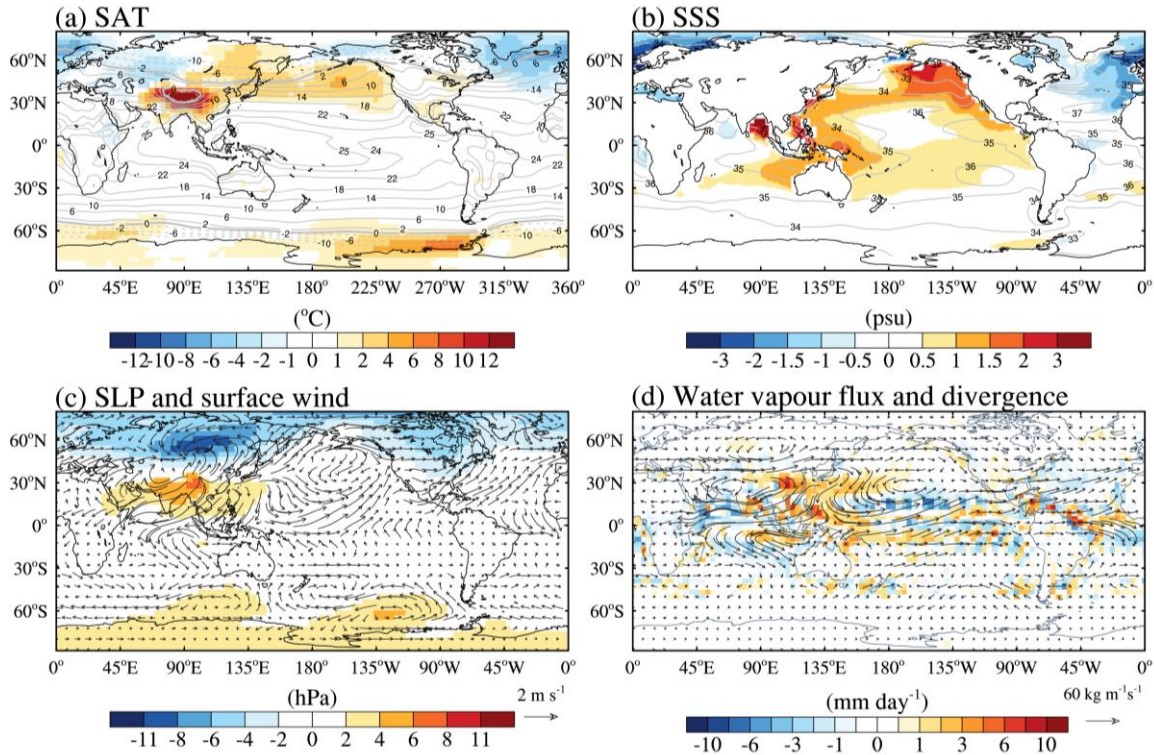

471

**Figure 2.** (a) Climatological SAT in MTP (contour) and anomalies (shaded) for NTP minus MTP;

(b) same as Figure 2a, but for sea surface salinity (SSS); (c) changes in sea-level pressure (SLP,

shading) and surface wind (vectors); and (d) vertically integrated (surface to 300 hPa pressure layer)

water vapor flux (vectors) and its convergence (shading) in NTP relative to MTP. Unit of

convergence is converted to mm day$^{-1}$ assuming the density of liquid water as 1 g cm$^{-3}$.

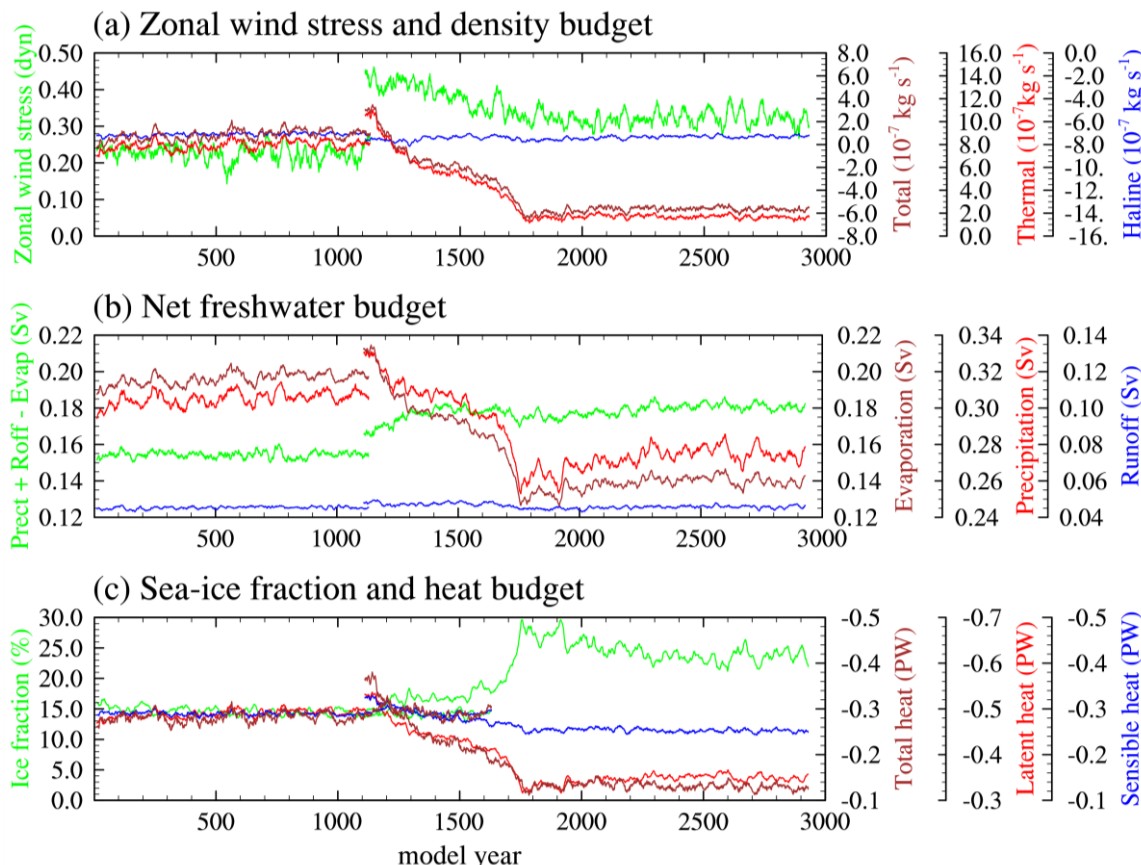

477

**Figure 3.** Regional annual mean across North Atlantic basin at 40 °–70 °N for MTP (1–1100 years) and NTP (1101–2940 years) of (a) zonal surface wind-stress, total density flux, haline density flux, and thermal density flux (total density flux is decomposed into haline contribution due to freshwater flux and thermal contribution due to heat flux (*Schmitt et al.*, 1989); (b) net freshwater, precipitation, runoff, and evaporation fluxes; (c) sea-ice fraction, total heat, sensible heat, and latent heat fluxes, (units: PW, 1 PW = $10^{15}$ W). For comparison purposes, all lines with common units use identical vertical scale spacing.

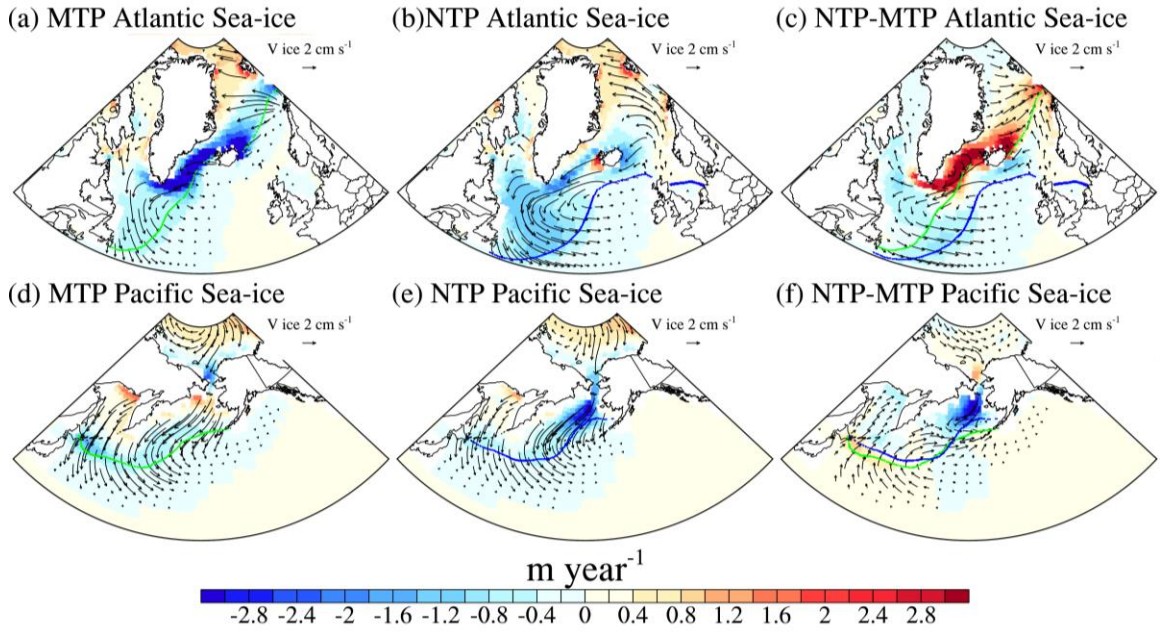

**Figure 4.** The North Atlantic and Pacific region features of annual mean sea-ice formation rate (shading; positive stands for formation, and negative stands for melting), sea-ice velocity (vectors, cm s$^{-1}$), and for (a, d) MTP, (b, e) NTP, and (c, f) difference between NTP and MTP. The February sea-ice margin is indicated with dashed lines (defined as the 15% sea-ice coverage, green line for MTP, blue line for NTP)

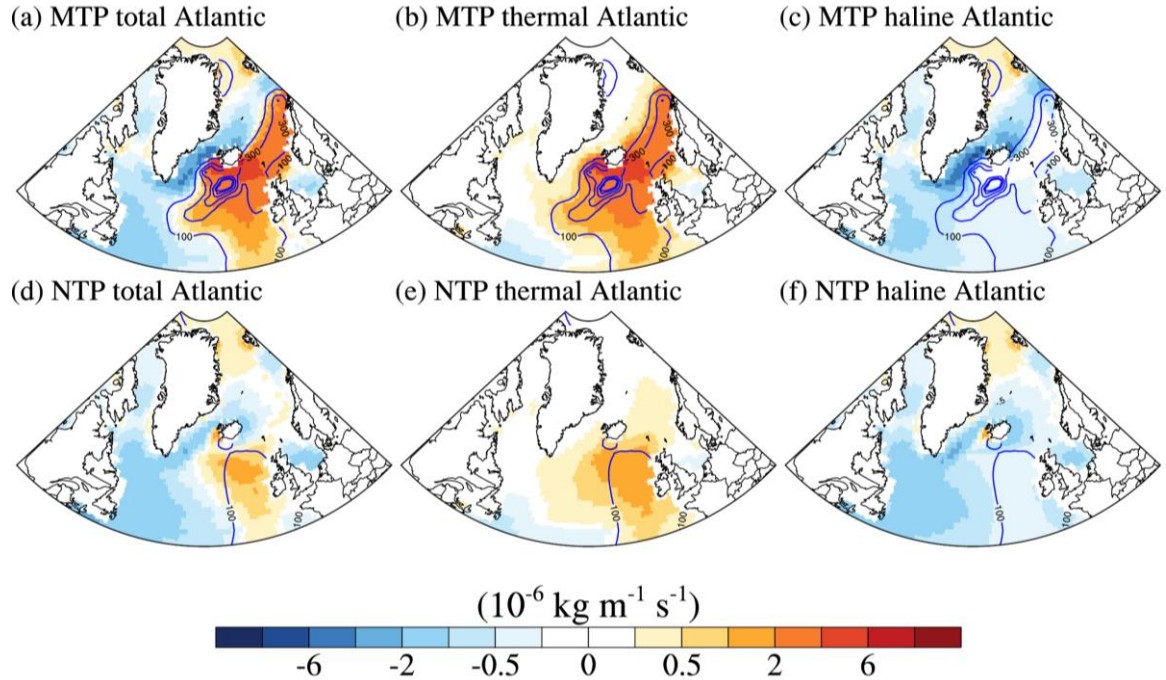

**Figure 5.** The North Atlantic annual mean (a, d) total density flux (shading; positive means flux makes water denser), (b, e) the thermal density flux, (c, f) the haline density flux, and the winter mixed layer depth (blue contour, contour interval: 200 m) in the MTP (upper panel) and NTP (lower panel).

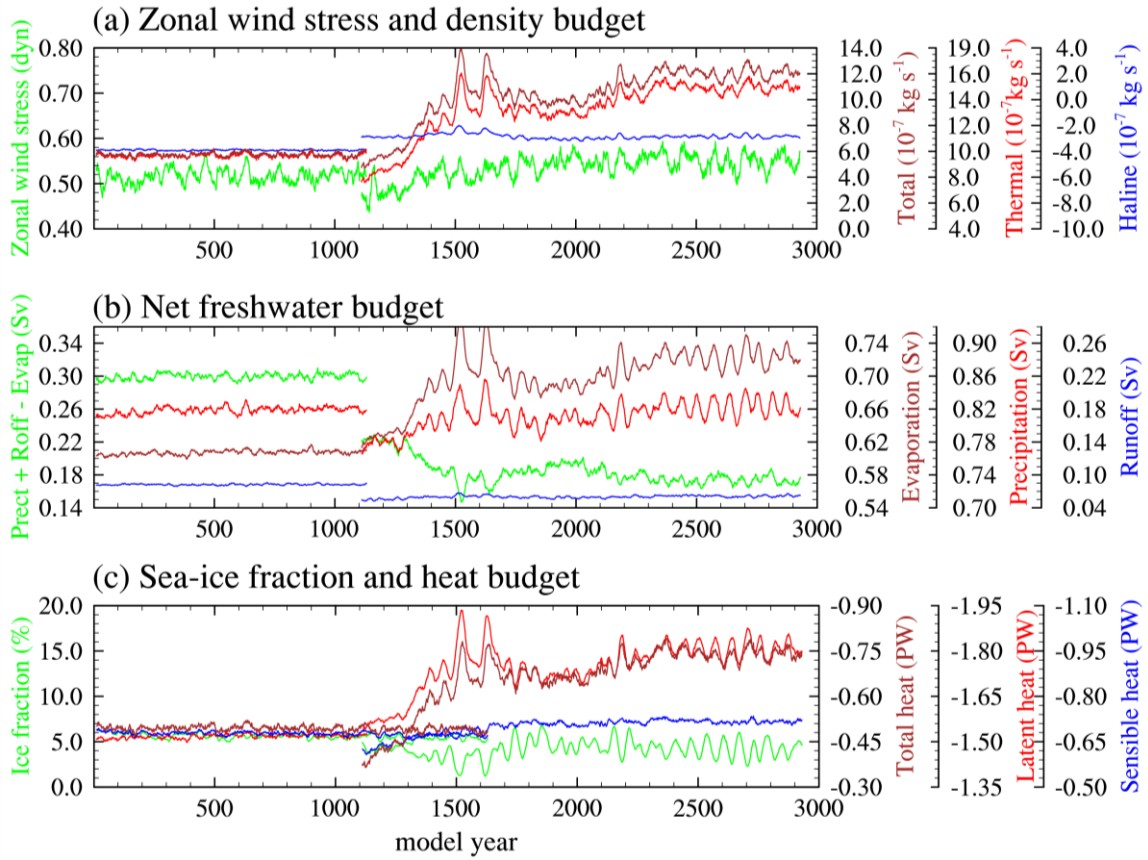


**Figure 6.** As in Figure 3, but for North Pacific basin at 30 °–70 °N.

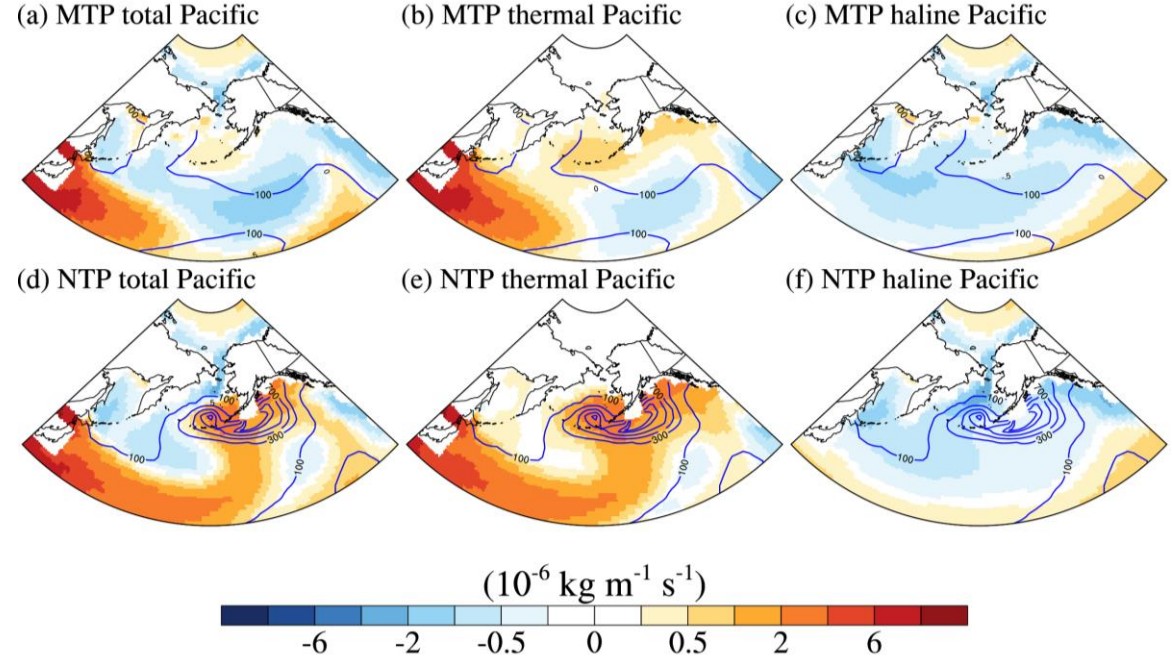


**Figure 7.** The same as Figure 5, but for the North Pacific (30–80 °N).

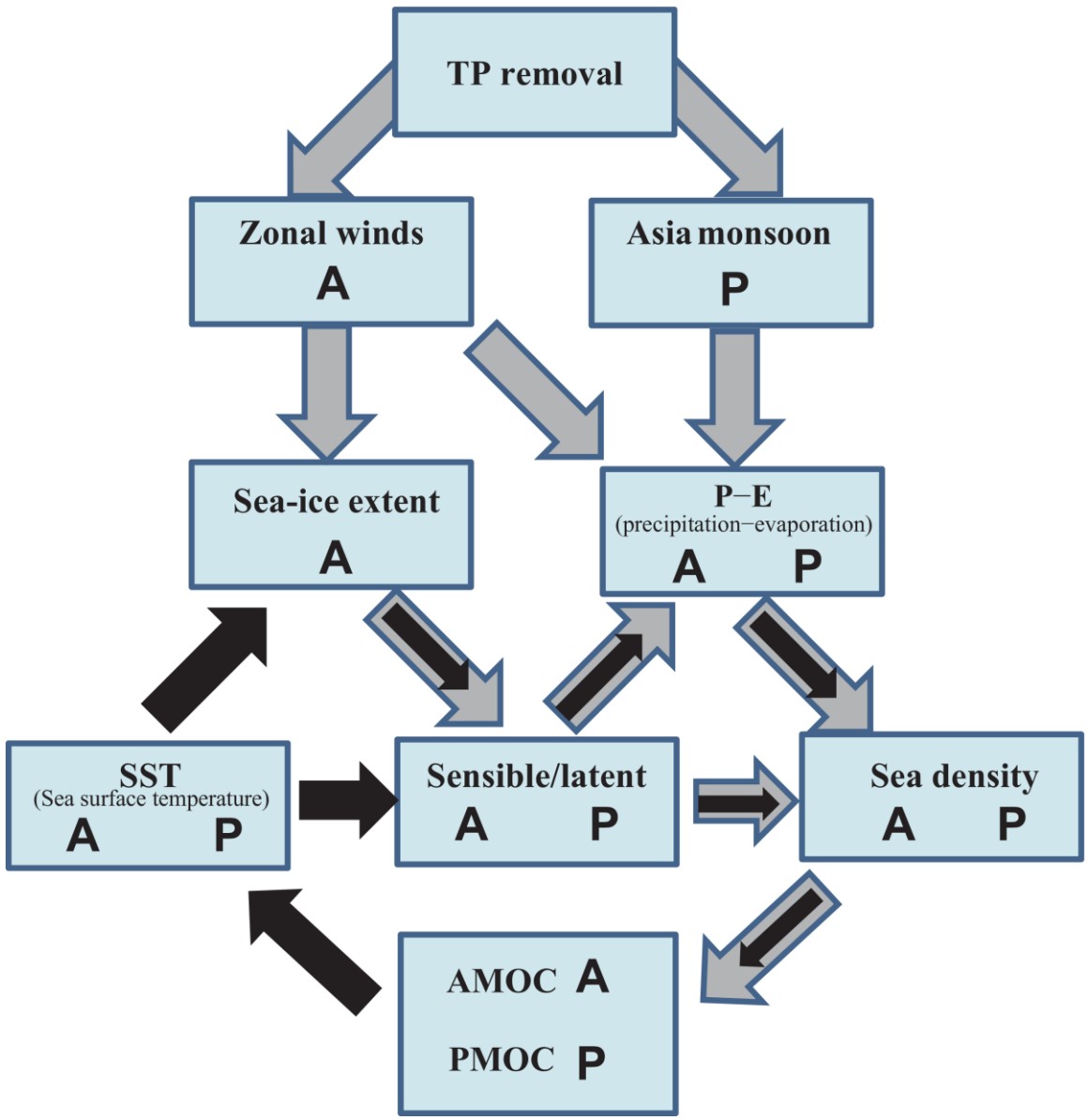


**Figure 8.** Schematic diagram about the influence of the removal of TP on the AMOC and PMOC. Vectors in gray denote the climate responses in relation to the increased in wind-induced and decreased monsoonal-driven net precipitation-evaporation and wind-driven sea-ice processes. The black color vectors denote the feedback processes. The bold characters A and P stand for the physical processes occurring over the North Atlantic and Pacific, respectively.