# Peer review of "overturning circulation in response to the uplift of the Tibetan Plateau"

_Climate of the Past, 2017_

## Referee Comment (RC1) · Anonymous Referee #1 · 26 Oct 2017

In the paper, Su and co-authors demonstrate that the uplift of TP leads to the weakening of AMOC, but the intensification of PMOC. The paper is well written. If the authors can address or discuss the two fundamental weaknesses below, I think the revised version will be largely improved.

1) Why removal of TP leads to the intensified westerlies over the North Atlantic and weakened subtropical anticyclones and trade winds over the Pacific? Are these changes a direct atmospheric response of TP removal, or including feedbacks of SST? In the revised version, the authors should explain why these atmospheric changes appear in their simulations. This explanation is helpful to judge if the weakening of AMOC

is an amplified result due to feedbacks of ocean circulations and SST.

2) In the NTP experiment, the net freshwater flux increases by 0.005 Sv at the initial stage, but by 0.025 Sv at the final stage. The author should discuss a little bit in the revised version, if fresh water flux of 0.005Sv is strong enough to trigger the weakening of AMOC. Is the model too sensitivity to a small change in fresh water flux?
* * *

---

## Referee Comment (RC2) · Anonymous Referee #2 · 11 Feb 2018

This is a fairly straightforward and worthwhile study of how Tibetan uplift may have affected global climate. The authors describe a logical pair of GCM experiments to diagnose the role of the Tibetan Plateau (TP) as it could have influenced the climate during the Cenozoic, although these simulations do not account for changes in other boundary conditions such as CO2 and continental configuration. The paper does a good job of providing motivation for this study, noting that most attention to mountain uplift on climate has focused on atmospheric dynamics, rather than ocean circulation (especially in high latitudes). My overall impression of the paper is favorable and that it is worthy of publication, subject to several mostly minor issues explained below.

Major Comments:

1. Using a low-resolution GCM is probably necessary for the long simulations needed for this study, and the authors do a commendable job on page 12 of discussing possible limitations of the low resolution on their conclusions. However, the T31 version of CESM that is used in this paper is known to have significant climate biases, especially in high-latitude regions that are a main focus for this paper (including Arctic sea ice extent). For the global ocean, these biases include a long-term drift in volumetric temperature and salinity, as suggested in Figure 1c. Implications of these model biases on the results and conclusions of this study are warranted.

Minor Comments:

1. The text contains many minor grammatical errors involving the usage of articles (i. e., when to use "a" or "the" before a noun). A thorough proofreading should cure this problem.

2. For readers not familiar with the geologic history of Tibetan Plateau uplift, please cite upfront the timing of this evolution. Line 59 of the Introduction lists a vague mention of "Given the timing of TP uplift. . . ", but it does not specify when that occurred. Only in the Conclusions section are relevant dates revealed.

3. Lines 63-65: I'm not completely clear of the reasoning implied here. Are the authors saying that the required integration time of their model simulations is so long that it's impractical to test additional parameters besides topography? Please clarify.

4. Lines 113-114: Given the importance of AMOC in these results, it would help to elaborate on how the model compares with the observed strength. What is the best observational estimate, including the range? Also, how does the simulated 6 Sv strength of PMOC in the reference simulation compare with observations?

5. Figure 1: Very interesting how the strength of AMOC and PMOC flip almost exactly between the two experiments, such that one or the other is around 18 Sv in MTP and

[Figure]

NTP. Is that a coincidence, or does it reflect a meridional heat transport requirement that is met by either ocean basin in the two different climates?

6. Lines 132-134: Why does a warmer Tibet reduce the Eurasia-Pacific thermal contrast during summer? Figure 2a indicates a much warmer Tibetan region and an overall warmer Asian landmass. Likewise, the next paragraph describes an associated weaker monsoon circulation, but that also seems counterintuitive with a much warmer Tibet. For example, many studies show that excessive cold (warmth) resulting from abnormally high (low) snow cover on the plateau is associated with a weaker (stronger) summer monsoon.

7. Line 161: why use the term "on the other hand" when describing the role of increased sea ice coverage? That implies a contrast with the previous sentence, which reports an increase in freshwater flux over the Atlantic. Yet both expanded sea ice extent and greater freshwater flux cause a lower surface density and thus favor a weakened AMOC.

8. Line 260: Which hypothesis is being referred to here—the one about the MOC being determined by large mountains or the one about asymmetric continental extents and basin geometries between the Atlantic and Pacific basins?

9. Lines 274-276: Why would planetary cooling during the Cenozoic lead to a reduced equator-to-pole thermal gradient? Colder global climates usually have even larger cooling in polar regions, giving rise to the term "polar amplification".

10. Page 13: Good observational evidence for a stronger PMOC during the early Cenozoic to support the model results presented here.

11. Page 14 and elsewhere: The authors rightfully point out that they have only tested the direct role of Tibetan topography and therefore ignored possible coinciding influences of other boundary conditions, such as higher greenhouse gas concentrations that may be relevant for the actual paleoclimatic conditions resulting from Tibetan uplift. I recall a paper by Vavrus and Kutzbach (2002, GRL) that involved a similar modeling study, but it isolated the individual impacts of mountain uplift and higher CO2 on AMOC. That article might be relevant for the present study.

---

## Author Comment (AC1) · 21 Mar 2018

Response to Reviewer #1:

In the paper, Su and co-authors demonstrate that the uplift of TP leads to the weakening of AMOC, but the intensification of PMOC. The paper is well written. If the authors can address or discuss the two fundamental weaknesses below, I think the revised version will be largely improved.

(1) Why removal of TP leads to the intensified westerlies over the North Atlantic and weakened subtropical anticyclones and trade winds over the Pacific? Are these changes a direct atmospheric response of TP removal, or including feedbacks of SST? In the revised version, the authors should explain why these atmospheric changes appear in their simulations. This explanation is helpful to judge if the weakening of AMOC is an amplified result due to feedbacks of ocean circulations and SST.

First, the intensified westerlies over the North Atlantic in response to the TP removal is due to the absence of barrier effect from the TP allowing the atmospheric jet stream and associated low-level winds to become more zonal. Besides, the standing waves in association with the TP removal are also absent, leading to less orographic gravity wave drag and stronger winds (Palmer et al., 1986; Sinha et al., 2012). As shown in the revised version, the intensified surface westerly winds over the upstream of TP are clearly simulated (Fig. 2c on page 22).

Meanwhile, both theoretical studies and numerical simulations have demonstrated that the TP uplift strengthens the Asian summer monsoon and in turn diabatic heating over the Asian monsoon region, which provides a critical role in the formation and maintenance of the summer North Pacific subtropical high (Ruddiman and Kutzbach, 1989; Rodwell and Hoskins, 2001; Kitoh, 2004). In particular, during the summertime, the upper-level divergence associated with strong monsoonal ascending motion is located over the Asian region and adjacent oceans, while the upper level convergence circulation related to the descending motion is observed over the middle and lower latitudes of eastern Pacific (Please see the following Fig. S1a).

Comparatively, this circulation cell in the NTP is not as strong as in the MTP, indicating the weakening of both the North Pacific subtropical high and Asian monsoon (Please see the following Figs. S1b and S1c).

[Figure]

Figure S1. The Northern Hemisphere summertime mean velocity potential (contour) and divergent wind (vectors) at 200 hPa for the (a) MTP, (b) NTP, and (c) the differences between NTP and MTP.

Second, following your comment, we have supplemented a set of experiments with and without TP undertaken by the atmospheric general circulation model (AGCM) of CESM

version 1.0.5, aiming to examine the feedback originating from the sea surface temperature changes. The 200-year climatologically averaged sea surface temperature prescribed in both the AGCM experiments is obtained from the MTP experiments undertaken by coupled atmosphere–ocean general circulation model (CGCM, CESM version 1.0.5). Both AGCM experiments are integrated for 50 model years, and the further analysis is performed based on the results of the last 30-year simulations.

In the CGCM experiments, the removal of the TP significantly causes a weakening of the North Pacific subtropical high and an overall weakening of the low-level tropical trade winds (Please see the following Fig. S2a). Similar changes are seen in the AGCM experiments, but with an overall weaker intensity (Please see the following Fig. S2b). Relative to the results of AGCM experiments, there is a clear decrease of tropical trade winds over the North Pacific and an increase of low-level westerly over the North Atlantic in the CGCM experiments (Please see the following Fig. S2c), indicating that the changes of sea surface temperature and oceanic circulations further amplify atmospheric circulation anomalies due to the TP uplift. Thus, we believe that the decrease of AMOC in our simulations is primarily attributed to the direct response of atmospheric circulation to the removal of the TP, and the oceanic feedback further amplifies this response. Considering the aforementioned atmospheric responses in association with the TP uplift have been addressed by many AGCM and CGCM simulations, we do not describe them in detail. Alternatively, we have added a short description on these processes in the revised version (L149–154).

[Figure]

Figure S2. Changes of annual mean 500 hPa vertical velocity and 850 hPa wind in NTP relative to MTP for the (a) CGCM, (b) AGCM, and (c) differences between CGCM and AGCM. Green contours surround the areas with topography higher than 2000 m and 4000 m, respectively.

(2) In the NTP experiment, the net freshwater flux increases by 0.005 Sv at the initial stage, but by 0.025 Sv at the final stage. The author should discuss a little bit in the revised version, if fresh water flux of 0.005Sv is strong enough to trigger the weakening of AMOC. Is the model too sensitivity to a small change in fresh water flux?

We have compared the magnitude of net freshwater flux in our simulations with two

earlier simulations, although there exists difference in the experimental configuration. Sinha et al. (2012) indicated a final net precipitation flux anomaly of approximately 0.02 Sv across the Atlantic basin (30–60 ˚N). Maffre et al. (2017) indicated that the net freshwater flux anomaly over the North Atlantic (22–60 ˚N) is approximately 0.0446 Sv and 0.097 Sv at the beginning and final stages, respectively. First, it should be noted that these two previous simulations with respect to the freshwater flux anomaly are obtained from simulations with and without global mountains, rather than the TP alone in our experiments. As such, the atmospheric circulation responses to the topographic modification in their experiments might be stronger than ours. Second, the area that they calculated the freshwater flux is over the North Atlantic at 22–60 ˚N, and the grid mesh is greater in size than ours at 40–70 ˚N. Third, Maffre et al. (2017) emphasized the net freshwater flux over the tropical Atlantic could play an important role in the AMOC weakening. We do not exclude the importance of the net freshwater flux over the tropical–subtropical Atlantic, since it has been well revealed that the warm and salty waters of the tropical–subtropical North Atlantic circulate north to the sub-polar regions of the North Atlantic via the Gulf Stream, and evaporation causes the surface waters to cool and thus the formation of North Atlantic Deep Water. Even though the net freshwater flux in our simulations is less than that in Maffre et al. (2017), it is difficult for us to determine the sensitivity of AMOC in response to the net freshwater flux in our simulations. Because we have identified the important role of wind-driven sea-ice process in initially triggering the AMOC weakening in our simulations. Thus, to answer the abovementioned question, a series of sensitivity experiments with the modification of the net freshwater flux are necessary to be performed. However, this topic is beyond the scope of this study and needs to be investigated in the future.

Reference

Kitoh, A.: Effects of mountain uplift on East Asian summer climate investigated by a coupled atmosphere–ocean GCM, J. Clim., 17, 783–802, 2004.

Maffre, P., Ladant, J. B., Donnadieu, Y., Sepulchre, P., and Goddéris, Y.: The influence of orography on modern ocean circulation, Clim. Dyn., doi:10.1007/s00382-017-3683-0, 2017.

Palmer, T. N., Shutts, G. J., and Swinbank, R.: Alleviation of a systematic westerly bias in general circulation and numerical weather prediction models through an orographic gravity wave drag parametrization, Q. J. R. Meteorol. Soc., 112, 1001–1039, 1986.

Rodwell, M. J. and Hoskins, B. J.: Subtropical amticyclones and summer monsoons, J. Clim., 14, 3192–3211, 2001.

Ruddiman, W. F., and Kutzbach, J. E.: Forcing of Late Cenozoic northern hemisphere climate by plateau uplift in southern Asia and the American West, J. Geophys. Res., 94(D15), 18409–18427, 1989.

Sinha, B., Blaker, A. T., Hirschi, J., Bonham, S., Brand, M., Josey, S., Smith, R. S., and Marotzke, J.: Mountain ranges favour vigorous Atlantic meridional overturning, Geophys. Res. Lett., 39, L02705, doi:10.1029/2011GL050485, 2012.

---

## Author Comment (AC2) · 21 Mar 2018

**Response to Reviewer #2:**

This is a fairly straightforward and worthwhile study of how Tibetan uplift may have affected global climate. The authors describe a logical pair of GCM experiments to diagnose the role of the Tibetan Plateau (TP) as it could have influenced the climate during the Cenozoic, although these simulations do not account for changes in other boundary conditions such as CO2 and continental configuration. The paper does a good job of providing motivation for this study, noting that most attention to mountain uplift on climate has focused on atmospheric dynamics, rather than ocean circulation (especially in high latitudes). My overall impression of the paper is favorable and that it is worthy of publication, subject to several mostly minor issues explained below.

We sincerely acknowledge you for very careful reading of the paper, including appropriate comments and useful suggestions. We try to answer your different suggestions and modify the paper accordingly.

Major Comments:

1. Using a low-resolution GCM is probably necessary for the long simulations needed for this study, and the authors do a commendable job on page 12 of discussing possible limitations of the low resolution on their conclusions. However, the T31 version of CESM that is used in this paper is known to have significant climate biases, especially in high-latitude regions that are a main focus for this paper (including Arctic sea ice extent). For the global ocean, these biases include a long-term drift in volumetric temperature and salinity, as suggested in Figure 1c. Implications of these model biases on the results and conclusions of this study are warranted.

We agree that the low resolution version of CESM has a cold bias against the observation, especially in the North Atlantic high latitudes, which is partly attributed to the deficit of ocean heat transport and the excess of Arctic sea-ice (Shields et al., 2012). More specifically, in the pre-industrial simulation (MTP), there is a weak positive slope in the long-term global mean

temperature, in particular at the beginning of simulations; and global mean temperature reaches a quasi-equilibrium state at approximately 12 ℃, which is lower than observations (please see Fig. 1c on page 21). In the revised version, we have mentioned this model bias and its potential influence on the simulations (L257–259).

Minor comments

1. The text contains many minor grammatical errors involving the usage of articles (i. e., when to use "a" or "the" before a noun). A thorough proofreading should cure this problem.

We have tried our best to improve the English writing in the revised version.

2. For readers not familiar with the geologic history of Tibetan Plateau uplift, please cite upfront the timing of this evolution. Line 59 of the Introduction lists a vague mention of "Given the timing of TP uplift. . . ", but it does not specify when that occurred. Only in the Conclusions section are relevant dates revealed.

The suggested revision has been made in the revised version (P3L59–62): In addition, it is suggested that the regional surface of the TP had reached a high elevation of more than 4000 meters around 40 Ma ago (Dupont-Nivet et al., 2008; Wang et al., 2008), although debates regarding paleoaltitude reconstructions remain (Botsyun et al., 2016).

3. Lines 63-65: I'm not completely clear of the reasoning implied here. Are the authors saying that the required integration time of their model simulations is so long that it's impractical to test additional parameters besides topography? Please clarify.

We are sorry that we do not express our meaning clearly. Actually, we want to say that the numerical experiments with and without the TP can be performed under different boundary conditions, such as high atmospheric $CO_2$ concentration and different orbital parameter configurations, rather than under the present-day boundary conditions. However, that kind of

simulations need to run the coupled model for a long time to reach a quasi-equilibrium state under those boundary conditions for the first step, and then the further experiments with and without the TP are carried out under specified question-dependent boundary conditions. For this reason, here we restrict our analysis to the sensitivity experiment of the TP uplift only under the present-day boundary conditions.

Considering this statement has been given in the section of "Model and experiments" (L87–89), that sentence has been removed from the revised version.

4. Lines 113-114: Given the importance of AMOC in these results, it would help to elaborate on how the model compares with the observed strength. What is the best observational estimate, including the range? Also, how does the simulated 6 Sv strength of PMOC in the reference simulation compare with observations?

Based on the continuous observations of AMOC from the RAPID (Rapid Climate Change) mooring array beginning in 2004 at 26.5°N Atlantic section, the intensity of AMOC is estimated to be 18.7 ±5.6 Sv for 2004–2005 (Cunningham et al., 2007) and approximately 17.5 Sv for the average during the period 2004–2012 (Smeed et al., 2014), respectively. We have added the related information in the revised version. On the other side, we do not provide PMOC-related estimation for the present-day, because there is not available observation on the thermohaline circulation in the North Pacific.

5. Figure 1: Very interesting how the strength of AMOC and PMOC flip almost exactly between the two experiments, such that one or the other is around 18 Sv in MTP and NTP. Is that a coincidence, or does it reflect a meridional heat transport requirement that is met by either ocean basin in the two different climates?

We have tried to discuss the potential relationship between the strength of AMOC and PMOC based on previous simulations. Hu et al. (2010) indicated that the establishment of the

PMOC but the collapse of the AMOC during the glacial times could be achieved only when the Bering Strait was closed, and this is because the Bering Strait played an important role in controlling the freshwater exchange between the Atlantic and Pacific. A recent modeling study suggested that the atmospheric moisture transport changes, in response to the greatly reduced meridional sea surface temperature gradients during the Pliocene, were capable of eroding the halocline, leading to the formation of the PMOC (Burls et al., 2017). More recently, Hutchinson et al. (2018) performed a numerical experiment, adopting the realistic boundary conditions at the Eocene–Oligocene Transition, to exhibit a bipolar sinking in the North Pacific and Southern Ocean, and, at the same time, they showed that the North Atlantic salinities are too fresh to permit sinking, due to surface transport from the fresh Arctic. In addition, it is simulated that the PMOC strength declines approximately linearly with mean salinity until it reaches a small background value similar to the present-day ocean (Cael and Ferrari, 2017).

Taking all these pertinent studies into account, it implies that the strength of PMOC is closely linked to the comprehensive inter-basin contrast between the North Pacific and the North Atlantic, including the freshwater budget, sea water density, and thermal differences. Of course, it is worth noting that the simulated global oceanic heat transport was little affected by the collapse of the AMOC in response to the global mountains, because of the heat compensation effect from the establishment of the PMOC (Maffre et al., 2017). To a certain degree, that does indicate a meridional heat transport requirement that is met by either ocean basin in the two different climates as you suggested. This issue has been beyond the scope of this study, and more studies are needed to address the related questions.

6. Lines 132-134: Why does a warmer Tibet reduce the Eurasia-Pacific thermal contrast during summer? Figure 2a indicates a much warmer Tibetan region and an overall warmer

Asian landmass. Likewise, the next paragraph describes an associated weaker monsoon circulation, but that also seems counterintuitive with a much warmer Tibet. For example, many studies show that excessive cold (warmth) resulting from abnormally high (low) snow cover on the plateau is associated with a weaker (stronger) summer monsoon.

As you mentioned, relative to the MTP experiment, the simulated strong warming occurs over the TP and East Asian continent in the NTP experiment (Please see Fig. 2a on page 22). This warming mainly reflects the effect of the atmospheric lapse rate, that is, the atmospheric temperature decreases with elevation. However, strong cooling occurs over the TP and the surrounding continent but slight change occurs over the mid-latitude North Pacific at both 850 hPa and 500 hPa (Fig. S3), leading to the decrease of zonal Eurasia–Pacific thermal contrast in the middle and lower troposphere and thus the weakening of subtropical anticyclones and trade winds over the North Pacific as indicated by Ruddiman and Kutzbach (1989), Rodwell and Hoskins (2001), and Kitoh (2004). The related sentence has been revised accordingly (L137–140). In response to the above large-scale circulation anomalies as shown in Fig. 2c, the removal of the TP leads to a significant divergence of moisture over East Asia and the western North Pacific marginal seas (Please see Fig. 2d on page 22), which is linked to a weakened monsoon circulation and is consistent with the previous simulations using both atmospheric and coupled models (Liu and Yin, 2002; Kitoh, 2004; Molnar et al., 2010). By the way, the studies you mentioned focus on the effect of the snow cover on the TP, and the related mechanism may differ from that of the removal of the TP in our simulations.

[Figure]

Figure S3. The (a) 850 hPa and (b) 500 hPa annual mean air temperature anomalies between NTP and MTP. In panel (a), the regions with an elevation higher than 1500 meters are left blank.

7. Line 161: why use the term "on the other hand" when describing the role of increased sea ice coverage? That implies a contrast with the previous sentence, which reports an increase in freshwater flux over the Atlantic. Yet both expanded sea ice extent and greater freshwater flux cause a lower surface density and thus favor a weakened AMOC.

"on the other hand" has been changed to "Moreover" in the revised version (L169).

8. Line 260: Which hypothesis is being referred to here—the one about the MOC being determined by large mountains or the one about asymmetric continental extents and basin geometries between the Atlantic and Pacific basins?

The original sentence "Our simulations support this hypothesis and highlight the significant role of the TP alone in supporting the modern AMOC" has been revised to "Our simulations support the hypothesis proposed by Warren (1983) and highlight the significant role of the TP alone in maintaining the modern AMOC" accordingly (L270–L272).

9. Lines 274-276: Why would planetary cooling during the Cenozoic lead to a reduced equator-to-pole thermal gradient? Colder global climates usually have even larger cooling

in polar regions, giving rise to the term "polar amplification".

We agree that the planetary cooling during the Cenozoic would lead to an increased rather than a reduced equator-to-pole thermal gradient. The relevant sentence has been revised as follows: The Earth has experienced a long-term cooling trend throughout the Cenozoic as testified by many proxies and stacked records (Zachos et al., 2001, 2008), in association with a increased equator-to-pole thermal gradient (L286–288).

10. Page 13: Good observational evidence for a stronger PMOC during the early Cenozoic to support the model results presented here.

Thanks for your appreciation.

11. Page 14 and elsewhere: The authors rightfully point out that they have only tested the direct role of Tibetan topography and therefore ignored possible coinciding influences of other boundary conditions, such as higher greenhouse gas concentrations that may be relevant for the actual paleoclimatic conditions resulting from Tibetan uplift. I recall a paper by Vavrus and Kutzbach (2002, GRL) that involved a similar modeling study, but it isolated the individual impacts of mountain uplift and higher CO2 on AMOC. That article might be relevant for the present study.

Thanks for your information. Vavrus and Kutzbach (2002) has been added in the revised version accordingly (L315).

References:

Burls, N. J., Fedorov, A. V., Sigman, D. M., Jaccard, S. L., Tiedemann, R., and Haug, G. H.: Active Pacific meridional overturning circulation (PMOC) during the warm Pliocene. Sci. Adv., 3, e1700156, 2017.

Cael, B. B., and Ferrari, R.: The ocean's saltiness and its overturning, Geophys. Res. Lett., 44, 1886–1891, doi:10.1002/2016GL072223, 2017.

Cunningham, S. A., Kanzow, T., Rayner, D., Baringer, M. O., Johns, W. E., Marotzke, J., Longworth, H. R., Grant, E. M., Hirschi, J. J.-M., Beal, L. M., Meinen, C. S., and Bryden, H. L.: Temporal variability of the Atlantic meridional overturning circulation at 26.5 °N, Science, 317, 935–938, 2007.

Gent, P. R., Danabasoglu, G., Donner, L. J., Holland, M. M., Hunke, E. C., Jayne, S. R., Lawrence, D. M., Neale, R. B., Rasch, P. J., Vertenstein, M., Worley, P. H., Yang, Z. L., and Zhang, M.: The Community Climate System Model version 4, J. Clim., 24, 4973–4991, doi:10.1175/2011JCLI4083.1, 2011.

Hu, A., G. Meehl, A., Han, W., Abe-Ouchi, A., Morrill, C., Okazaki, Y., and Chikamoto, M. O.: The Pacific-Atlantic seesaw and the Bering Strait, Geophys. Res. Lett., 39, L03702, doi:10.1029/2011GL050567, 2012.

Hutchinson, D. K., de Boer, A. M., Coxall, H. K., Caballero, R., Nilsson, J., and Baatsen, M.: Climate sensitivity and meridional overturning circulation in the late Eocene using GFDL CM2.1, Clim. Past Discuss., https://doi.org/10.5194/cp-2017-161, in review, 2018.

Kitoh, A.: Effects of mountain uplift on East Asian summer climate investigated by a coupled atmosphere–ocean GCM, J. Clim., 17, 783–802, 2004.

Liu, X., and Yin, Z.-Y.: Sensitivity of East Asian monsoon climate to the uplift of the Tibetan Plateau, Palaeogeogr. Palaeoclim. Palaeoecol., 183, 223–245, 2002.

Maffre, P., Ladant, J. B., Donnadieu, Y., Sepulchre, P., and Goddéris, Y.: The influence of orography on modern ocean circulation, Clim. Dyn., doi:10.1007/s00382-017-3683-0, 2017.

Molnar, P., Boos, W. R., and Battisti, D. S.: Orographic controls on climate and paleoclimate of Asia: Thermal and mechanical roles for the Tibetan Plateau, Annu. Rev. Earth. Planet.

Sci., 38, 77–102, 2010.

Rodwell, M. J., and Hoskins, B. J.: Subtropical anticyclones and summer monsoons, J. Clim., 14, 3192–3211, 2001.

Ruddiman, W. F., and Kutzbach, J. E.: Forcing of Late Cenozoic northern hemisphere climate by plateau uplift in southern Asia and the American West, J. Geophys. Res., 94(D15), 18409–18427, 1989.

Shields, C. A., Bailey, D. A., Danabasoglu, G., Jochum, M., Kiehl, J. T., Levis, S., and Park, S.: The low-resolution CCSM4, J. Clim., 25, 3993–4014, 2012.

Smeed, D. A., McCarthy, G. D., Cunningham, S. A., Frajka-Williams, E., Rayner, D., Johns, W. E., Meinen, C. S., Baringer, M.O., Moat, B. I., Duchez, A., Bryden, H. L.: Observed decline of the Atlantic Meridional Overturning Circulation 2004–2012, Ocean Sci., 10, 29–38, 2016.

Vavrus, S. and Kutzbach, J. E.: Sensitivity of the thermohaline circulation to increased $CO_2$ and lowered topography, Geophys. Res. Lett., 29, 1546, doi:10.1029/2002GL014814, 2002.